# The Influence of Weight Loss in Postural Control in Women Undergoing Sleeve Gastrectomy

**DOI:** 10.3390/jfmk7040117

**Published:** 2022-12-19

**Authors:** Valentina Bullo, Stefano Gobbo, Lucia Cugusi, Andrea Di Blasio, Alessandro Bortoletto, Roberto Pippi, David Cruz-Diaz, Andrea Gasperetti, Roberto Vettor, Andrea Ermolao, Marco Bergamin

**Affiliations:** 1Department of Medicine, University of Padova, Via Giustiniani 2, 35128 Padova, Italy; 2Department of Biomedical Sciences, University of Sassari, Viale San Pietro 43/B, 07100 Sassari, Italy; 3Department of Medicine and Sciences of Aging, G. D’Annunzio University of Chieti-Pescara, Via dei Vestini 31, 66100 Chieti, Italy; 4Healthy Lifestyle Institute, C.U.R.I.A.Mo. (Centro Universitario Ricerca Interdipartimentale Attività Motoria), Department of Medicine and Surgery, University of Perugia, Via G. Bambagioni, 19, 06126 Perugia, Italy; 5Department of Health Sciences, Faculty of Health Sciences, University of Jaén, 23071 Jean, Spain; 6Department of Medicine, Sports and Exercise Medicine Division, University of Padova, Via Giustiniani 2, 35128 Padova, Italy; 7Clinica Medica 3, Department of Medicine, Azienda Ospedaliera Padova, University of Padova, 35122 Padova, Italy

**Keywords:** postural control, bariatric surgery, obesity

## Abstract

Background: Bariatric surgery is the most effective procedure for obesity management, with a greater body weight loss and the remission of several diseases. The aim of this study was to analyze the relationships between the anthropometric profile and postural control outcomes in a group of obese adult women, and the effect of bariatric surgery on postural control. Methods: eighty-eight women candidates for bariatric surgery were recruited. Static balance was measured with the ARGO stabilometric platform under two conditions: open eyes (OE) and closed eyes (CE). Results: Multiple linear regression indicated BMI as the first predictor for postural control in all parameters, except for APO in open eyes, predicted mainly by height. Changes in body weight and BMI showed no statistically significant correlations with modification of postural control parameters (OE), while they appeared to exert an influence under closed eyes conditions. Conclusions: Before surgery, obese patients with a higher BMI showed a better postural control. After surgery, the sway path and antero-posterior oscillation improved under open eyes conditions, while the magnitude of weight loss was negatively correlated with differences in postural control.

## 1. Introduction

Obesity is a chronic, progressive, and treatable multifactorial and neurobehavioral pathology affecting 650 million adults worldwide [1] and caused by environmental, behavioral, and genetic factors [2,3,4]. Obesity has a negative impact on general health, quality of life, and an increased risk of falling into disability and morbidity [5] with adverse metabolic [6], biomechanical, [7,8] and psychosocial health consequences. Moreover, epidemiological studies have shown an increased risk of falls related to obesity [9,10,11,12], due to alterations of postural control patterns [13,14]. Non-surgical management of obesity consists of a multicomponent approach, including behavioral therapy, pharmacotherapies, and lifestyle change, with the aim to reduce energy intake with diet, increase physical activity [15] and reduce sedentary behavior [16,17]. Because few patients achieve an important weight loss with a lifestyle change, many undergo bariatric surgery [15]. From an epidemiological point of view, obese individuals are distributed equally between the two sexes [1]. However, gender distribution among obese patients who undergo bariatric surgery is approximately 80% of women and 30% of men [18].

Although bariatric surgery leads to a reduced fat mass, muscles and bone mass, which constitute fat-free mass (FFM), are also affected. Indeed, after bariatric surgery, a loss of FFM accounted for about 31% of weight loss [8,19], with the possibility of a negative effect on muscle strength, functional capacity, and physical performance (i.e., gait speed, time to rise from a chair, etc.). Absolute strength tends to decrease [20], while the relative strength and physical activity level tend to increase [20,21,22]. Moreover, static balance seems to improve [23]. Some authors reported that physical performance improves after bariatric surgery, but it is not clear whether the improvements are linked to body composition changes or to an increase in physical activity [24]. However, exercise, nutrition, and surgeons’ follow-up are often recommended after bariatric surgery [25,26]; nevertheless, dropouts are very frequent. Particularly, post-bariatric surgery exercise counseling sessions (including a resistance-training component) may help to improve muscle strength, which is related to sarcopenic obesity, functional capacity, and all-cause mortality risk [27].

Bariatric surgery causes a rapid weight loss affecting postural control [28,29]; the effects of rapid weight loss after bariatric surgery are not yet clearly explained, especially on the locomotor system, motor control, and postural stability [29]. Postural stability is commonly evaluated by the center of pressure (CoP) [30]. Previous research suggests that an increased body fat mass decreases postural stability in obese older men based on increased CoP velocity [14,31]. Furthermore, in people with obesity, the maintenance of postural balance and body stability are more difficult during walking and position changes [29], because they have a reduced ability to control sways due to a lower relative muscle strength than healthy-weight people [32]. As mentioned above, functional and physiological changes in musculoskeletal composition, the center of gravity, and coordination are caused by the loss of muscle mass induced by bariatric surgery. This aspect could compromise stability [22]. Some authors suggest that stability increases in obese subjects after significant weight loss, observing a strong linear relationship between the magnitude of the weight loss and an improvement in balance control [33].

In light of these perspectives, the aim of this study was to analyze the relationships between the anthropometric profile and postural control outcomes in a group of obese adult women, and the effect of bariatric surgery on postural control. The research hypotheses are, firstly, that the amplitude of oscillations increases with increasing body weight and, secondly, that there is an improvement in postural control correlated with the amount of weight loss due to bariatric surgery.

## 2. Materials and Methods

### 2.1. Participants

Eighty-eight obese women, candidates for bariatric surgery, were recruited from the Sport and Exercise Medicine Division of the (* blind to the reviewer *). Inclusion criteria were: (a) BMI > 35; (b) undergoing Sleeve Gastrectomy surgery within 1 month from the evaluation; (c) no previous bariatric surgery; (d) ability to speak or understand the Italian language; (e) mini-mental state examination higher than 26. Exclusion criteria were: (a) chronic conditions that could influence postural control (e.g., multiple sclerosis, history of cancer, etc.); (b) an uncompleted functional evaluation; (c) other techniques of bariatric surgery different from sleeve gastrectomy. Subjects who met these criteria were informed about our study purpose and gave written consent for participation. The investigation complied with the current laws of Italy for research on human participants and was approved by the University Hospital Board n. 2027 dated 12 of January 2017. Baseline characteristics are reported in Table 1.

### 2.2. Medical Examination and Postural Control Evaluation

Participants’ height and weight were measured with a stadiometer (Ayrton Corporation, Model S100, Prior Lake, MN, USA) and an electronic scale (Home Health Care Digital Scale, Model MC-660, C-7300, MO, USA), on the day of assessment. A medical examination, cardiopulmonary exercise test, and mini-mental state examination were administered to all participants at the first visit (before surgery). The mini-mental state examination was used to identify cognitive impairment [34] and to exclude subjects with a result lower than 26. Static balance was measured with the ARGO stabilometric platform (RGMD, Genoa, Italy). The evaluation of static balance was performed under two conditions: with open and closed eyes, with two trials each. Participants with visual impairments performed the test with their daily glasses or contact lenses. In both tests, subjects were required to stand in an upright position as still as possible, with their feet together and their arms at their sides. During the test with eyes open, the subject had to stare at a reference point located on the blackboard for 30 s. During the test with eyes closed, the subject had to stay on the platform for 30 s with closed eyes. In both trials, four parameters were recorded: Sway Path (SP), Sway Area (SA), Anterior-posterior oscillation (APO), and Medio-lateral oscillation (MLO). These measures were collected at a 100 Hz sampling rate. Each patient performed the same test protocol 1 month before and 6 months after sleeve gastrectomy (SG) surgery in random order.

### 2.3. Statistical Analysis

Statistical analyses were conducted with SPSS (Version 21.0 for Windows, SPSS Inc., Chicago, IL, USA). Data are presented as a mean ± standard deviation. A Shapiro–Wilk test was applied to check the normal distribution of all the variables. A comparison between pre- and post-SG was performed with the paired t-test for normally distributed variables; otherwise, the Wilcoxon–Mann–Whitney test was performed. Hierarchical stepwise multiple regression was conducted between pre-surgery postural control outcomes and anthropometrics parameters (age, weight, height, and BMI). Correlation coefficients were calculated between independent variables to determine the level of collinearity. Only body weight and BMI showed a high and significant correlation (ρ = 0.76, *p* < 0.001), so body weight was not considered for the model. Pearson’s correlation coefficient (ρ) was calculated between pre- and post-surgery changes on postural control outcomes and weight loss (Δ = post-pre). A value of *p* < 0.05 was considered statistically significant. The effect size (ES) of each outcome measure was calculated following the formula: ES = (mean pre-value − mean post value)/SD pre-value. Interpretation was performed according to Cohen [35] and Sawilowsky’s guidelines [36]. Sample size calculation was based on the mean values of mediolateral oscillation detected in a previous study [37]. To this end, the following equation was applied: N = (2(SD2)) × (Zα + Zβ)2)/Δ2.

## 3. Results

Height and weight were used to calculate the body mass index (BMI). Anthropometric and medical status modifications are reported in Table 2.

No comorbidities were reported by 20.5% of participants. Pre-diabetes and type 2 diabetes mellitus were present in 14.8% of women. Hypertension was present in 36.4%, obstructive sleep apnea syndrome affected 5.7% of the sample, while dyslipidemia was diagnosed in 26.1% of women. Finally, 20.5% of women had hypothyroidism. After bariatric surgery, significant reductions in body weight (−29.7 kg, −26.7%, ES = 2.2, *p* < 0.001) and BMI (−11.4 kg/m^2^, −26.6%, ES = 2.4, *p* < 0.001) were found (Table 2).

Static balance was evaluated with a stabilometric platform under two different conditions: eyes opened (EO) and eyes closed (EC). After SG, postural control improved with a significant decrease of SP (−1.2 mm/s, ES = 0.3, *p* < 0.001) and APO (−2.9 mm^2^/s, ES = 0.4, *p* < 0.05), while static balance evaluated with eyes closed showed no significant changes. All data are reported in Table 3.

### 3.1. Pre-Surgery Anthropometric Characteristics and Postural Control Outcomes

Figure 1 shows the Pearson correlation between BMI and postural control. Table 4 shows the linear multiple regression between pre-surgery postural control outcomes and anthropometric characteristics. Under open eyes conditions, BMI is the first predictive value for SP (8.3%, *p* < 0.05), SA (10.1%, *p* < 0.05), and MLO (14.1%, *p* < 0.001), while height is the first predictor for APO (8.2%, *p* < 0.05). Moreover, BMI is the first predictor for all parameters under closed eyes conditions. In detail, BMI accounted for 6.4% of the variance of SP (*p* < 0.05), 11.4% of SA (*p* = 0.001), 6.3% of APO (*p* < 0.05), and 15% of MLO (*p* < 0.001).

### 3.2. Post-Surgery Modification and Postural Control Changes

Figure 2 and Figure 3 show Pearson’s correlation coefficient between post-surgery modifications and postural control changes. Under open eyes conditions, changes in body weight and BMI showed no statistically significant correlations with modifications of postural control parameters. On the contrary, postural control under closed eyes conditions seems to be influenced by body weight and BMI reduction. In detail, weight loss was negatively correlated with differences of SP (ρ = −0.27, CI = [−0.45; −0.06], *p* = 0.012), SA (ρ = −0.25, CI = [−0.44; −0.04], *p* = 0.018), APO (ρ = −0.41, CI = [−0.57; −0.22], *p* < 0.001), and MLO (ρ = −0.32, CI = [−0.5; −0.12], *p* = 0.002). Likewise, BMI loss was negatively correlated with differences of SP (ρ = −0.24, CI = [−0.43; −0.03], *p* = 0.024), SA (ρ = −0.25, CI = [−0.43; −0.04], *p* = 0.02), APO (ρ = −0.41, CI = [−0.57; −0.22], *p* < 0.001), and MLO (ρ = −0.32, CI = [−0.49; −0.12], *p* = 0.003).

## 4. Discussion

The aim of this study was to analyze the relationships between the anthropometric profile and postural control outcomes in a group of obese adult women, and the effect of bariatric surgery on postural control. Our results showed a significant negative correlation between BMI and postural oscillation before surgery, such as the positive correlation between weight loss and the modification of stabilometric parameters.

It is well established that obesity implies functional impairments [12]. However, the question of how obesity affects postural control is not yet well clarified. In our analyses, all models showed a significant correlation between anthropometric and postural parameters, indicating that the BMI, more than height and age, influences postural control. In detail, it seems that women with a higher BMI had a better postural control. This result is partially confirmed by the literature. Indeed, when comparing oscillation between obese and normal-weight women, obesity seemed to induce a better postural control with less APO [38,39] and MLO [37,38] than for normal-weight women. On the contrary, several studies found a significant correlation between body weight and postural control, indicating a worsening of balance with an increase in body weight [31,40,41,42]. One hypothesis that may explain these results is the base of support: subjects with obesity tend to be wider due to a high thigh circumference, potentially inducing an external deviation of the tibia and causing valgus knees [43]. This modification can be enhanced by the body distribution of adipose tissue, which may lead to a gynoid or android shape. A recent study [44] compared postural control of women with android and gynoid fat distribution, where CoP and APO velocities were higher in android women. Biomechanically, this result could be explained by the inverse pendulum model, where the greater distance between the center of mass (abdomen) and fulcrum (ankle joint) results in a lower postural stability [45]. This specific assumption can be declined with androgenic obesity, which requires a greater muscle effort at the ankle level [45]. Given that our sample consisted only of women, we can assume that the group was characterized by a gynoid distribution of body fat. Thus, as the weight increases, the postural control improves.

Meanwhile, bariatric surgery is the best strategy for fast weight loss, with health benefits [46]. After bariatric surgery, a loss of fat free mass (FFM) accounted for about 31% of weight loss [19], with modifications of the functional capacity. Furthermore, the modifications induced by weight loss on postural control are not so clear. In this study, women showed a general improvement in postural control under open eyes conditions after SG, even if only the sway path and antero-posterior oscillation significantly improved. These results are partially confirmed by the literature; postural control in a group of obese men improved significantly after weight loss induced by surgery or a hypocaloric diet [23,33], while another study found no significant modifications in postural control in a group of obese individuals undergoing bariatric surgery [29].

Another point of interest in this study is the magnitude analysis of weight loss and pre- to post-surgery postural control modifications. To the best of our knowledge, only Teasdale and colleagues evaluated the relationship between the change in body weight and the change in CoP velocity in men [33], finding a linear reduction in CoP speed with an increasing weight loss. In our analysis, postural control is negatively correlated with weight loss magnitude only under closed eyes conditions. We can hypothesize that the neuro-muscular and/or proprioceptive component of balance could be influenced by the results. In fact, obesity is associated with a lower plantar sensitivity [47], which seems to be associated with a poor postural control [48]. Future research is needed to analyze in depth the relationships between the magnitude of weight loss and modifications on postural control, while in particular integrating other health outcomes and functional capacities, such as comorbidities, drugs, and muscular strength.

This study has several limitations. Firstly, body circumferences may be measured to better understand adipose tissue distribution and its influence on postural control. Secondly, muscular strength evaluations could be integrated to analyze their interaction with postural control. Thirdly, the subjects included were younger than 60 years old and potentially without a postural control deficit resulting from aging. Finally, the physical activity level has not been evaluated to determine if it influences postural control before and after surgery. Future research could include an integrated assessment protocol of the functional capacity in obese subjects, especially from the viewpoint of exercise prescription. Additionally, multidisciplinary intervention including specific balance training induces postural control improvement in subjects with obesity [49], suggesting the need to integrate specific exercises after surgery, especially in the elderly [50].

## 5. Conclusions

Postural control in obese women is influenced by BMI. In detail, women with a higher BMI showed better postural control, and weight loss induced by surgery improved it. However, under closed eyes conditions, postural control is negatively correlated with weight loss magnitude, indicating possible balance deficits in patients who have had a large weight loss. This finding underlines the importance of assessing balance in these patients in order to integrate balance training to prevent any balance deficits. Future research is needed to understand the more in-depth modification of postural control after surgery, also including specific balance training integration.

## Figures and Tables

**Figure 1 jfmk-07-00117-f001:**
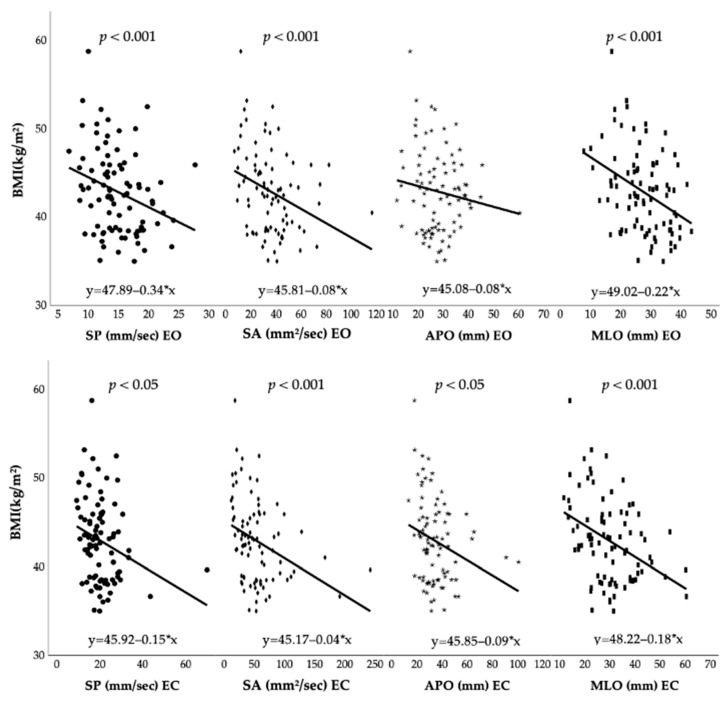
Pearson correlation between pre-surgery BMI and postural control outcomes. Abbreviations: BMI: Body Mass Index; SP: sway path; SA: sway area; APO: antero-posterior oscillation; MLO: medio-lateral oscillation; EO: eyes opened; EC: eyes closed.

**Figure 2 jfmk-07-00117-f002:**
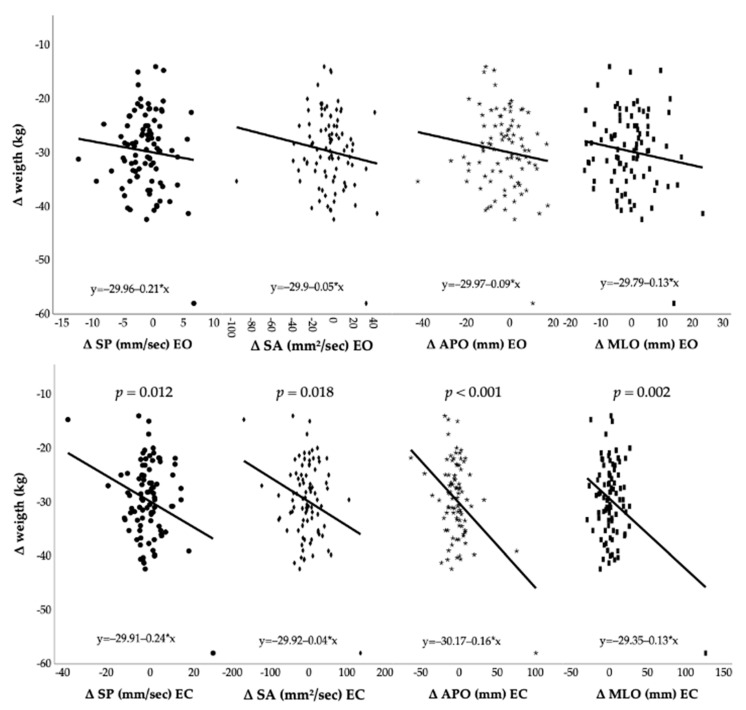
Post-surgery weight loss and postural control changes. Abbreviations: Δ: difference between post- and pre-surgery outcome; SP: sway path; SA: sway area; APO: antero-posterior oscillation; MLO: medio-lateral oscillation; EO: eyes opened; EC: eyes closed.

**Figure 3 jfmk-07-00117-f003:**
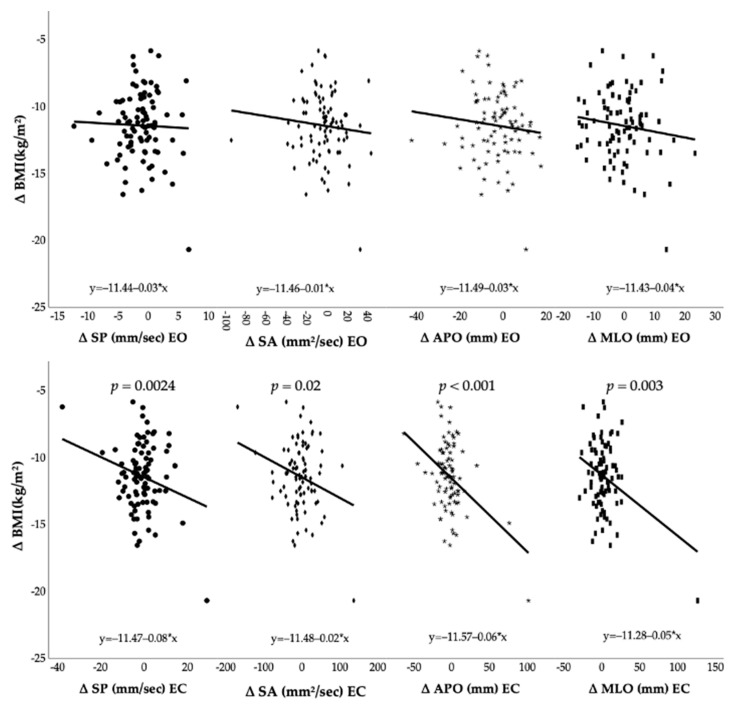
Post-surgery BMI loss and postural control changes. Abbreviations: Δ: difference between post- and pre-surgery outcome; BMI: Body Mass Index; SP: sway path; SA: sway area; APO: antero-posterior oscillation; MLO: medio-lateral oscillation; EO: eyes opened; EC: eyes closed.

**Table 1 jfmk-07-00117-t001:** Baseline participants’ characteristics before sleeve gastrectomy surgery (mean ± standard deviation).

Characteristic	Mean ± SD	Range (Max-Min)
Sex (n)	Women (88)	
Age (years)	44.4 ± 11.2	18–68
Height (m)	1.6 ± 0.1	1.4–1.75
Weight (kg)	111.4 ± 13.5	82.5–140
BMI (kg/m^2^)	42.9 ± 4.7	35–58.77
MMSE (score)	29.2 ± 1	26–30
Days from surgery to post-test	189.36 ± 14.6	126–248
Comorbidities (type)	Pre-diabetes (13), DMT2 (13), hypothyroidism (18), dyslipidemia (23), IPTS (32), OSAS (5), musculoskeletal disorders (27), other (34)	
Comorbidities (num)	No com. (18), 1 com. (20), 2 com. (18), 3 com. (16), 4 com. (10), 5 com. (5), >6 com. (1)	
Drugs (num)	No drug (26), 1 drug (23), 2 drugs (13), 3 drugs (7), 4 drugs (5), 5 drugs (6), >6 drugs (8)	
Obesity class (%)	II obesity class: 28 (33%) III obesity class: 59 (67%)	

Abbreviation: BMI: body mass index; WC: waist circumference; MMSE: mini-mental state examination; II class obesity: BMI 35–39.9; III class obesity: BMI > 40; DMT2: type 2 diabetes mellitus; IPTS: hypertension; OSAS: obstructive sleep apnea syndrome; num: number; com: comorbidities.

**Table 2 jfmk-07-00117-t002:** Changes in anthropometric and medical status parameters after SG.

	Pre (m ± sd)	Post (m ± sd)	Δ (abs) C.I. (95%)	Δ (%)	ES
Weight (kg)	111.4 ± 13.5	81.7 ± 10.7 **	−29.7 [−31.2; −28.2]	−26.7%	2.2
BMI (kg/m^2^)	42.9 ± 4.7	31.5 ± 4.1 **	−11.4 [−11.9; −10.9]	−26.6%	2.4
Pre-diabetes (num)	13	7	−6		
DMT2 (num)	13	8	−5		
Hypothyroidism (num)	18	18	0		
Dyslipidemia (num)	23	17	−6		
IPTS (num)	32	19	−13		
OSAS (num)	5	3	−2		
MSDs (num)	27	21	−6		
Other (num)	34	28	−6		

** *p* < 0.001; Abbreviation: C.I.: confidence interval; BMI: body mass index; DMT2: type 2 diabetes mellitus; IPTS: hypertension; OSAS: obstructive sleep apnea syndrome; MSD: musculoskeletal disorders; num: number; m ± sd: mean ± standard deviation; Δ: absolute change from post to pre; ES: effect size.

**Table 3 jfmk-07-00117-t003:** Changes in anthropometric and postural control parameters after SG.

	Pre (M ± SD)	Post (M ± SD)	Δ (abs) C.I. (95%)	Δ (%)	ES
SP (mm/s) EO	14.8 ± 4	13.5 ± 3.4 **	−1.2 [−1.9; −0.6]	−8.2%	0.3
SA (mm^2^/s) EO	35.8 ± 18.4	31.9 ± 13.9	−4 [−8; 0.1]	−11%	0.2
APO (mm) EO	27.9 ± 8.2	25 ± 8 *	−2.9 [−5; −0.8]	−10.4%	0.4
MLO (mm) EO	27.4 ± 7.9	26.8 ± 6.2	−0.6 [−2.2; 1]	−2.2%	0.1
SP (mm/s) EC	20.5 ± 8.1	19.7 ± 7.2	−0.8 [−2.5; 0.9]	−4%	0.1
SA (mm^2^/s) EC	54 ± 37.9	49.2 ± 31	−4.8 [−13.4; 3.9]	−8.8%	0.1
APO (mm) EC	34.6 ± 13.9	31.7 ± 15.5	−2.9 [−6.9; 1.1]	−8.4%	0.2
MLO (mm) EC	30 ± 10.3	32.7 ± 15.3	+2.8 [−1; 6.5]	9.2%	−0.3

* *p* < 0.05; ** *p* < 0.001; Abbreviation: C.I.: confidence interval; SP: sway path; SA: sway area; APO: antero-posterior oscillation; MLO: medio-lateral oscillation; EO: eyes opened; EC: eyes closed; M ± SD: mean ± standard deviation; Δ: absolute pre-post change; ES: effect size.

**Table 4 jfmk-07-00117-t004:** Linear multiple regression between pre-surgery anthropometric characteristics and postural control outcomes.

Postural Parameters	Model	R	R^2^	Adjusted R^2^	F	*p* Value
SP (mm/s) EO	(1) BMI	0.289	0.083	0.073	7.814	0.006
	(2) BMI, age	0.324	0.105	0.084	4.970	0.009
	(3) BMI, age, height	0.330	0.109	0.077	3.413	0.021
SA (mm^2^/s) EO	(1) BMI	0.317	0.101	0.09	9.615	0.003
	(2) BMI, height	0.333	0.111	0.09	5.305	0.007
	(3) BMI, height, age	0.351	0.123	0.092	3.945	0.011
APO (mm) EO	(1) Height	0.286	0.082	0.071	7.651	0.007
	(2) Height, BMI	0.298	0.089	0.067	4.127	0.019
	(3) Height, BMI, age	0.315	0.099	0.067	3.093	0.031
MLO (mm) EO	(1) BMI	0.376	0.141	0.131	14.153	<0.001
	(2) BMI, age	0.384	0.147	0.127	7.350	0.001
	(3) BMI, age, height	0.391	0.153	0.122	5.046	0.003
SP (mm/s) EC	(1) BMI	0.253	0.064	0.053	5.905	0.017
	(2) BMI, age	0.320	0.102	0.081	4.844	0.010
	(3) BMI, age, height	0.323	0.104	0.072	3.251	0.026
SA (mm^2^/s) EC	(1) BMI	0.338	0.114	0.104	11.074	0.001
	(2) BMI, age	0.367	0.135	0.114	6.613	0.002
	(3) BMI, age, height	0.369	0.136	0.105	4.406	0.006
APO (mm) EC	(1) BMI	0.252	0.063	0.053	5.824	0.018
	(2) BMI, height	0.274	0.075	0.053	3.442	0.037
	(3) BMI, height, age	0.306	0.094	0.061	2.900	0.040
MLO (mm) EC	(1) BMI	0.387	0.150	0.140	15.145	<0.001
	(2) BMI, height	0.390	0.152	0.132	7.615	<0.001
	(3) BMI, height, age	0.399	0.159	0.129	5.294	0.002

Abbreviations: R: correlation coefficient; R^2^: multiple correlation coefficient; BMI: body mass index; SP: sway path; SA: sway area; APO: antero-posterior oscillation; MLO: medio-lateral oscillation; EO: eyes opened; EC: eyes closed.

## Data Availability

The data presented in this study are available on request from the corresponding author. The data are not publicly available due to privacy.

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
