# Peer review of "The Influence of Weight Loss in Postural Control in Women Undergoing Sleeve Gastrectomy"

_jfmk, 2022, doi:10.3390/jfmk7040117_

Round 1
Reviewer 1 Report
The authors present an original paper investigating the effects of weight loss induced by surgery in women on postural stability. This is an area that has received a little attention in the literature, therefore, warrants further examination. Overall, the manuscript is written and organized fairly well. It follows the logical sequence of a research purpose. Despite this strength, I have few comments that need to be addressed by the authors and listed below.
TITLE
Title does not represent the overall results of the study. It seems that the presented results pointed out that postural stability is increased after surgery and weight loss (see results comments section)
ABSTRACT.
Abstract should be modified according to my comments in results comments section.
INTRODUCTION.
Since the objective of the study is related to weight loss after surgery and postural control, first and second paragraphs should be merged and more synthetized.
Otherwise, paragraphs 3 to 5 present properly key published studies related to the purpose of the current study.
METHODS
Table 1 should be referenced in 2.1 Participants and should presented physical participants’ characteristics before and after surgery.
Section 2.2 should be named “medical examination and postural control evaluation” since physical activity quantification is not presented in the result section.
Equipment used should be properly referenced: 1) Stadiometer (Model S100; Ayrton Corp., Prior Lake, MN, USA), 2) electronic scale (Home Health Care Digital Scale, Model MC-660, C-7300, MO, USA) and 3) Static balance was measured with a stabilometric platform (ARGO; RGMD SpA., Genoa, Italy).
Postural control evaluation. How many trials per condition have been realized? Authors should indicate how the moments and force data have been digitally filtered before computing CP displacements.
2.3 Statistical analyses
Authors have enough subjects and variables to conduct linear multiple regression. The power of this analysis would be better than simple correlations to observe and predict the relationships between anthropometric characteristics and postural control variables before weight loss. Pearson’s correlation is fine to observe the relationships between the magnitude of weight loss and delta changes in postural control variables.
RESULTS
First sentence should be removed.
Tables 2 to 4 did not show a worsen postural control after weight loss. Table 2 point out a significant difference for SP and APO in EO. However, all delta% changes show a reduction of postural control values, meaning a diminution of the excursion and displacements of the center of pressure (postural stability is increased). It is not worsen!!!
Table 3 shows that, excepted for EO APO, postural control is related to BMI. Table 4 shows significant relationships between the amount of weight loss and changes in postural control parameters in EC.
Authors should present the relationship with a figure and not only the results in a table. A postural parameter should be choose to illustrate this relationship among all postural parameters used.
DISCUSSION
The discussion should be reorganized according to the results comments and new analyses suggested. Postural control seems to be better after weight loss and there is a relationship between the amount of weight loss and the increase of postural stability.
Author Response
Comments and Suggestions for Authors
The authors present an original paper investigating the effects of weight loss induced by surgery in women on postural stability. This is an area that has received a little attention in the literature, therefore, warrants further examination. Overall, the manuscript is written and organized fairly well. It follows the logical sequence of a research purpose. Despite this strength, I have few comments that need to be addressed by the authors and listed below.
- TITLE
Title does not represent the overall results of the study. It seems that the presented results pointed out that postural stability is increased after surgery and weight loss (see results comments section)
AUTHORS: we agree with reviewer. We change it. The influence of weight loss in postural control in women undergoing sleeve gastrectomy
- ABSTRACT
Abstract should be modified according to my comments in results comments section.
AUTHORS: we modify the abstract.
- INTRODUCTION
Since the objective of the study is related to weight loss after surgery and postural control, first and second paragraphs should be merged and more synthetized. Otherwise, paragraphs 3 to 5 present properly key published studies related to the purpose of the current study.
AUTHORS: as suggest, we reduce the two paragraphs.
- METHODS
Table 1 should be referenced in 2.1 Participants and should presented physical participants’ characteristics before and after surgery.
AUTHORS: table 1 reported baseline characteristics of participants. We referenced it in paragraph 2.1. In results section we reported table 2 (new table) with modification after surgery.
Section 2.2 should be named “medical examination and postural control evaluation” since physical activity quantification is not presented in the result section.
AUTHORS: we agree and modify the name
Equipment used should be properly referenced: 1) Stadiometer (Model S100; Ayrton Corp., Prior Lake, MN, USA), 2) electronic scale (Home Health Care Digital Scale, Model MC-660, C-7300, MO, USA) and 3) Static balance was measured with a stabilometric platform (ARGO; RGMD SpA., Genoa, Italy).
AUTHORS: We are sorry, but we are not sure if we understand the question. Are the reviewer referring to the scientific validation of the instrumentation?
Postural control evaluation. How many trials per condition have been realized? Authors should indicate how the moments and force data have been digitally filtered before computing CP displacements.
AUTHORS: were performed two trials for each condition. These measures were collected at 100 Hz sampling rate; we filtered raw data using the ARGO software (RGDM, Genova, IT) that adopts a post-processing low-pass filtering with a 10 Hz frequency cutoff.
2.3 Statistical analyses. Authors have enough subjects and variables to conduct linear multiple regression. The power of this analysis would be better than simple correlations to observe and predict the relationships between anthropometric characteristics and postural control variables before weight loss. Pearson’s correlation is fine to observe the relationships between the magnitude of weight loss and delta changes in postural control variables.
AUTHORS: we integrated linear multiple regression
- RESULTS
First sentence should be removed.
AUTHORS: we remove it
Tables 2 to 4 did not show a worsen postural control after weight loss. Table 2 point out a significant difference for SP and APO in EO. However, all delta% changes show a reduction of postural control values, meaning a diminution of the excursion and displacements of the center of pressure (postural stability is increased). It is not worsen!!! Table 3 shows that, excepted for EO APO, postural control is related to BMI. Table 4 shows significant relationships between the amount of weight loss and changes in postural control parameters in EC.
AUTHORS: we change the title to better fit results. Weight loss improve postural control, but only in two parameters (table 3). However, comparing the difference between pre to post weight and pre to post oscillation results showed negative correlation. Some patients recorded an increase in oscillation after surgery. Comparing magnitude of weight loss with difference in pre- to post-surgery, higher is the weight loss and higher is the difference in oscillation (Table 5). With this result we hypothesize that subjects with higher weight loss might have postural control impairments.
Authors should present the relationship with a figure and not only the results in a table. A postural parameter should be choose to illustrate this relationship among all postural parameters used.
AUTHORS: as suggest by the reviewer, we include two figure to present the relationship between weight loss and changes in postural control (figure 1), and BMI changes and changes in postural control (figure 2)
- DISCUSSION
The discussion should be reorganized according to the results comments and new analyses suggested. Postural control seems to be better after weight loss and there is a relationship between the amount of weight loss and the increase of postural stability.
AUTHORS: we reorganize results according to previous comment and results.
Reviewer 2 Report
Thank you for the opportunity to review, I have the following comments on the text:
1. Introduction
1. Please add statistics on how many people are with obesity currently in the world. How many people have comorbidities? How much does it cost to treat people with obesity?
2. L56-59 - The sentence is not understandable, suggests rewording. So many women and men have obesity in the world, in the country ? What does this concern?
3. L62 - fat-free mass (FFM) - The shortcut is already developed a line above.
2. Materials and Methods
1. L96 – ‘’ (*blind to the reviewer*).’’, L104 ‘’ *** ‘’and 105 ‘’ ** (blind for review) ‘’ - Please insert the following information. Wants to review legislation in particular.
2. L100 - This is a rather extensive topic unfortunately abbreviated by the authors. Please explain who conducted the study ? Please elaborate on what other disease entities disqualified ? Were people with, for example, myopia disqualified ?
3. L111 - The test was conducted fasting or after eating ?
4. L111-114 - Please describe what the medical examination was like, what the mental status examination was like - what questionnaires?
5. L116-117 – ‘’ with open and closed eyes.’’ - Was there a random selection of the test ?
6. L116-117 – ‘’ with open and closed eyes’’ – Did the patients have a refractive defect (such as myopia) or not ? - According to recent studies, visual impairment and closed-eye and open-eye testing affect muscle changes. Please comment on this (doi: 10.3390/jcm10225376 and doi: 10.1038/s41598-022-13607-1).
7. L122 –‘’ Medio-lateral oscillation (MLO). ‘’ - According to whose methodology was this study ?
8. 2.3. Statistical analysis - How the sample size was calculated? / I would ask the authors to add a confidence interval.
4. Discussion
1. L222-224 - There is a lack of development on the topic of open eyes and the impact of this on attitude. Open eyes can cause increased muscle tension compared to closed eyes (doi: 10.3390/jcm10225376 and doi: 10.1038/s41598-022-13607-1). Increased muscle tension with a network of fascia (10.3390/ijms23105674), band of skeletal muscles or through the phenomenon of tensegration (10.1053/joca.1998.0164 ) will stabilize posture.
Please elaborate on this topic in the text.
5.Conclusions
1. In my opinion, the entire conclusions should be rewritten, there are too many repetitions of results and considerations. Examples of errors sentence L254-255 is a repetition of results. Sentence L257-258 is currently unconfirmed due to my methodological questions. After answering my methodological questions, I will be able to consider whether this is an acceptable sentence.
Author Response
Comments and Suggestions for Authors
Thank you for the opportunity to review, I have the following comments on the text:
- Introduction
Please add statistics on how many people are with obesity currently in the world. How many people have comorbidities? How much does it cost to treat people with obesity?
AUTHORS: we integrate the paragraph with the number of obese adults worldwide.
L56-59 - The sentence is not understandable, suggests rewording. So many women and men have obesity in the world, in the country? What does this concern?
AUTHORS: we reword the sentence. This concern is to justify our sample of only women.
L62 - fat-free mass (FFM) - The shortcut is already developed a line above.
AUTHORS: we delete the second shortcut
- Materials and Methods
L96 – ‘’ (*blind to the reviewer*).’’, L104 ‘’ *** ‘’and 105 ‘’ ** (blind for review) ‘’ - Please insert the following information. Wants to review legislation in particular.
AUTHORS: The investigation complied with the current laws of Italy for research on human participants and was approved by the University Hospital Board n. 2027 (blind for review) dated the 12 of January 2017.
L100 - This is a rather extensive topic unfortunately abbreviated by the authors. Please explain who conducted the study? Please elaborate on what other disease entities disqualified? Were people with, for example, myopia disqualified?
AUTHORS: the study was conducted by the Sport and Exercise Medicine Division of the University of Padova. Were disqualified subjects with history of cancer, multiple sclerosis, fibromyalgia, surgery of lower limb in the previous year. Were not excluded people with myopia, because the evaluations were performed using the optical glasses.
L111 - The test was conducted fasting or after eating?
AUTHORS: thanks for the sentence. The tests were performed on morning, among 9.00 to 13.00.
L111-114 - Please describe what the medical examination was like, what the mental status examination was like - what questionnaires?
AUTHORS: During the medical examination, were questioned about the presence of pathology, and drug therapy. Then blood pressure, heart rate, and resting VO2 were measured. Mini mental state examination was performed according to Folstein and colleagues, 1975
L116-117 – ‘’ with open and closed eyes.’’ - Was there a random selection of the test?
AUTHORS: yes, the tests were performed in random selection. We specify it at the end of the paragraph (These measures were collected at a 100 Hz sampling rate. Each patient performed the same test protocol 1 month before and 6 months after sleeve gastrectomy (SG) surgery in random order.).
L116-117 – ‘’ with open and closed eyes’’ – Did the patients have a refractive defect (such as myopia) or not? - According to recent studies, visual impairment and closed-eye and open-eye testing affect muscle changes. Please comment on this (doi: 10.3390/jcm10225376 and doi: 10.1038/s41598-022-13607-1).
AUTHORS: thank you for the question. Some patients had visual impairment, but patients wearing optical glasses or contact lenses performed the tests using them. The mentioned studies refer to alterations at the level of the masticatory and cervical muscles, without reference to postural control. If the reviewer can better explain what he/she meant to ask, let's try to answer.
L122 –‘’ Medio-lateral oscillation (MLO). ‘’ - According to whose methodology was this study?
AUTHORS: MLO is a stabilometric platform output, the maximal displacement of the CoP in medio-lateral directions (mm).
2.3. Statistical analysis - How the sample size was calculated? / I would ask the authors to add a confidence interval.
AUTHORS: Sample size calculation was based on the mean values of mediolateral oscillation detected in previous study (Kováčiková et al. 2014, doi: 10.5507/ag.2014.015). To this end the following equation was applied N=(2(SD2))*(Zα+Zβ)2 )/Δ2. SD is a standard deviation detected in a previous study based on the mean value of the mediolateral oscillation in obese group (SD= ±4); Zα is represented by α = 0.05 (1.96) and meaning the α point which a null hypothesis is rejected, 1- Zβ (80% power) representing the ability to detect a difference where it truly exists; Δ is the mean meaningful difference to determine an improvement in mediolateral oscillation. The net equation result is N = 32, as sub-group sample size. We add the interval confidence of correlation.
- Discussion
L222-224 - There is a lack of development on the topic of open eyes and the impact of this on attitude. Open eyes can cause increased muscle tension compared to closed eyes (doi: 10.3390/jcm10225376 and doi: 10.1038/s41598-022-13607-1). Increased muscle tension with a network of fascia (10.3390/ijms23105674), band of skeletal muscles or through the phenomenon of tensegration (10.1053/joca.1998.0164) will stabilize posture.
Please elaborate on this topic in the text.
AUTHORS: we are sorry, but in our opinion this topic is not relevant with our study. Open eyes induce impairment on functional indices (10.3390/jcm10225376) and an increased bioelectrical activity of cervical and masticatory muscles (10.1038/ s41598-022-13607-1) in subjects with myopia. However, the studies analyzed muscle activity in condition of rest, during maximal voluntary clenching in the intercuspal position, during maximal voluntary clenching in the intercuspal position on dental rollers, and maximal mouth opening. Moreover, evaluations were performed without visual correction. In our study, subject with visual impairments performed the test using their optical glasses or contact lenses to avoid alteration (we add this on medical examination and postural control evaluation paragraph).
4.Conclusions
In my opinion, the entire conclusions should be rewritten, there are too many repetitions of results and considerations. Examples of errors sentence L254-255 is a repetition of results. Sentence L257-258 is currently unconfirmed due to my methodological questions. After answering my methodological questions, I will be able to consider whether this is an acceptable sentence.
AUTHORS: we agree with the reviewer, and we rewrote the conclusion.
Round 2
Reviewer 1 Report
The authors answer adequatelty and properly to all my comments; I thank them for that, it is really appreciated. The manuscript have been significantly improved. Before recommandation for publication, few minor corrections need to be done.
1) Please, provide if possible, the mean time between surgery and post-test.
2) For all figures, please put the BMI on X-axis and postural variables on Y-axis. Please, insert in all graphs the equation of the the regression.
3) Paragraph 2.2 Medical examination. Please replace "Model S100; Ayrton Corp., Prior Lake, MN" by "Model S100; Ayrton Corp., Prior Lake, MN, USA" and replace "Home Health Care Digital Scale, Model MC-660, C-7300, MO" by "Home Health Care Digital Scale, Model MC-660, C-7300, MO, USA".
Author Response
The authors answer adequately and properly to all my comments; I thank them for that, it is really appreciated. The manuscript have been significantly improved. Before recommendation for publication, few minor corrections need to be done.
1) Please, provide if possible, the mean time between surgery and post-test.
AUTHORS: the mean time between surgery and post test was 189 days. We add this information on table 1
2) For all figures, please put the BMI on X-axis and postural variables on Y-axis. Please, insert in all graphs the equation of the the regression.
AUTHORS: thanks for the suggestion. We put BMI on X-asis and insert the equation of the regression in all figures.
3) Paragraph 2.2 Medical examination. Please replace "Model S100; Ayrton Corp., Prior Lake, MN" by "Model S100; Ayrton Corp., Prior Lake, MN, USA" and replace "Home Health Care Digital Scale, Model MC-660, C-7300, MO" by "Home Health Care Digital Scale, Model MC-660, C-7300, MO, USA".
AUTHORS: we follow the suggestion
Reviewer 2 Report
Thank you for sending your answers. Congrats on the amount of work you put into the manuscript.
Most of the corrections are acceptable. I have the following minor comments:
1. Please add information about sample size calculation in the text before publishing.
2. Please correct citations and biographies according to the publisher's style.
‘’In the text, reference numbers should be placed in square brackets [ ], and placed before the punctuation; for example [1], [1–3] or [1,3]. For embedded citations in the text with pagination, use both parentheses and brackets to indicate the reference number and page numbers; for example [5] (p. 10). or [6] (pp. 101–105).
The reference list should include the full title, as recommended by the ACS style guide. Style files for Endnote and Zotero are available.
References should be described as follows, depending on the type of work:
Journal Articles:
1. Author 1, A.B.; Author 2, C.D. Title of the article. Abbreviated Journal Name Year, Volume, page range’’
https://www.mdpi.com/journal/jfmk/instructions
‘’AUTHORS: thank you for the question. Some patients had visual impairment, but patients wearing optical glasses or contact lenses performed the tests using them. The mentioned studies refer to alterations at the level of the masticatory and cervical muscles, without reference to postural control. If the reviewer can better explain what he/she meant to ask, let's try to answer.’’
Information about the study with glasses and lenses is satisfactory to the reviewer.
Making the information more specific, it is impossible to separate changes in muscle chains or the fascial network from each other. If the subjects had been studied without correction it is likely that the entire muscular and fascial system could have been affected by this factor. The studys was given as an example (doi: 10.3390/jcm10225376 and doi: 10.1038/s41598-022-13607-1). This could affect research.
However, this mistake was not made.
Author Response
Thank you for sending your answers. Congrats on the amount of work you put into the manuscript.
Most of the corrections are acceptable. I have the following minor comments:
- Please add information about sample size calculation in the text before publishing.
AUTHORS: we add sample size calculation on statistic paragraph.
- Please correct citations and biographies according to the publisher's style.
‘’In the text, reference numbers should be placed in square brackets [ ], and placed before the punctuation; for example [1], [1–3] or [1,3]. For embedded citations in the text with pagination, use both parentheses and brackets to indicate the reference number and page numbers; for example [5] (p. 10). or [6] (pp. 101–105).
AUTHORS: we agree and correct the style.
The reference list should include the full title, as recommended by the ACS style guide. Style files for Endnote and Zotero are available.
References should be described as follows, depending on the type of work:
Journal Articles:
- Author 1, A.B.; Author 2, C.D. Title of the article. Abbreviated Journal Name Year, Volume, page range’’
https://www.mdpi.com/journal/jfmk/instructions